# On Poisson Graphical Models

**Eunho Yang**
Department of Computer Science
University of Texas at Austin
eunho@cs.utexas.edu

**Pradeep Ravikumar**
Department of Computer Science
University of Texas at Austin
pradeepr@cs.utexas.edu

**Genevera I. Allen**
Department of Statistics and
Electrical & Computer Engineering
Rice University
gallen@rice.edu

**Zhandong Liu**
Department of Pediatrics-Neurology
Baylor College of Medicine
zhandonl@bcm.edu

## Abstract

Undirected graphical models, such as Gaussian graphical models, Ising, and multinomial/categorical graphical models, are widely used in a variety of applications for modeling distributions over a large number of variables. These standard instances, however, are ill-suited to modeling count data, which are increasingly ubiquitous in big-data settings such as genomic sequencing data, user-ratings data, spatial incidence data, climate studies, and site visits. Existing classes of Poisson graphical models, which arise as the joint distributions that correspond to Poisson distributed node-conditional distributions, have a major drawback: they can only model negative conditional dependencies for reasons of normalizability given its infinite domain. In this paper, our objective is to modify the Poisson graphical model distribution so that it can capture a rich dependence structure between count-valued variables. We begin by discussing two strategies for truncating the Poisson distribution and show that only one of these leads to a valid joint distribution. While this model can accommodate a wider range of conditional dependencies, some limitations still remain. To address this, we investigate two additional novel variants of the Poisson distribution and their corresponding joint graphical model distributions. Our three novel approaches provide classes of Poisson-like graphical models that can capture both positive and negative conditional dependencies between count-valued variables. One can learn the graph structure of our models via penalized neighborhood selection, and we demonstrate the performance of our methods by learning simulated networks as well as a network from microRNA-sequencing data.

## 1   Introduction

Undirected graphical models, or Markov random fields (MRFs), are a popular class of statistical models for representing distributions over a large number of variables. These models have found wide applicability in many areas including genomics, neuroimaging, statistical physics, and spatial statistics. Popular instances of this class of models include Gaussian graphical models [1, 2, 3, 4], used for modeling continuous real-valued data, the Ising model [3, 5], used for modeling binary data, as well as multinomial graphical models [6] where each variable takes values in a small finite set. There has also been recent interest in non-parametric extensions of these models [7, 8, 9, 10]. None of these models however are best suited to model *count data*, where the variables take values in the set of all positive integers. Examples of such count data are increasingly ubiquitous in big-data

settings, including high-throughput genomic sequencing data, spatial incidence data, climate studies, user-ratings data, term-document counts, site visits, and crime and disease incidence reports.

In the univariate case, a popular choice for modeling count data is the Poisson distribution. Could we then model complex multivariate count data using some multivariate extension of the Poisson distribution? A line of work [11] has focused on log-linear models for count data in the context of contingency tables, however the number of parameters in these models grow exponentially with the number of variables and hence, these are not appropriate for high-dimensional regimes with large numbers of variables. Yet other approaches are based on indirect copula transforms [12], as well as multivariate Poisson distributions that do not have a closed, tractable form, and relying on limiting results [13]. Another important approach defines a multivariate Poisson distribution by modeling node variables as sums of independent Poisson variables [14, 15]. Since the sum of independent Poisson variables is Poisson as well, this construction yields Poisson marginal distributions. The resulting joint distribution, however, becomes intractable to characterize with even a few variables and moreover, can only model *positive correlations*, with further restrictions on the magnitude of these correlations. Other avenues for modeling multivariate count-data include hierarchical models commonly used in spatial statistics [16].

In a qualitatively different line of work, Besag [17] discusses a tractable and natural multivariate extension of the univariate Poisson distribution; while this work focused on the pairwise model case, Yang et al. [18, 19] extended this to the general graphical model setting. Their construction of a *Poisson graphical model* (PGM) is simple. Suppose all node-conditional distributions, the conditional distribution of a node conditioned on the rest of the nodes, are univariate Poisson. Then, there is a unique joint distribution consistent with these node-conditional distributions, and moreover this joint distribution is a graphical model distribution that factors according to a graph specified by the node-conditional distributions. While this graphical model seems like a good candidate to model multivariate count data, there is one major defect. For the density to be normalizable, the edge weights specifying the Poisson graphical model distribution have to be non-positive. This restriction implies that a Poisson graphical model distribution only models negative dependencies, or so called "competitive" relationships among variables. Thus, such a Poisson graphical model would have limited practical applicability in modeling more general multivariate count data [20, 21], with both positive and negative dependencies among the variables.

To address this major drawback of non-positive conditional dependencies of the Poisson MRF, Kaiser and Cressie [20], Griffith [21] have suggested the use of the *Winsorized* Poisson distribution. This is the univariate distribution obtained by truncating the integer-valued Poisson random variable at a finite constant $R$. Specifically, they propose the use of this Winsorized Poisson as node-conditional distributions, and assert that there exists a consistent joint distribution by following the construction of [17]. Interestingly, we will show that their result is incorrect and this approach *can never* lead to a consistent joint distribution in the vein of [17, 18, 19]. Thus, there currently does not exist a graphical model distribution for high-dimensional multivariate count data that does not suffer from severe deficiencies. In this paper, our objective is to specify a joint graphical model distribution over the set of non-negative integers that can capture rich dependence structures between variables.

The major contributions of our paper are summarized as follows: We first consider truncated Poisson distributions and (1) show that the approach of [20] is NOT conducive to specifying a joint graphical model distribution; instead, (2) we propose a novel truncation approach that yields a proper MRF distribution, the *Truncated PGM* (TPGM). This model however, still has certain limitations on the types of variables and dependencies that may be modeled, and we thus consider more fundamental modifications to the univariate Poisson density's base measure and sufficient statistics. (3) We will show that in order to have both positive and negative conditional dependencies, the requirements of normalizability are that the base measure of the Poisson density needs to scale quadratically for linear sufficient statistics. This leads to (4) a novel *Quadratic PGM* (QPGM) with linear sufficient statistics and its logical extension, (5) the *Sublinear PGM* (SPGM) with sub-linear sufficient statistics that permit sub-quadratic base measures. Our three novel approaches for the first time specify classes of joint graphical models for count data that permit rich dependence structures between variables. While the focus of this paper is model specification, we also illustrate how our models can be used to learn the network structure from iid samples of high-dimensional multivariate count data via neighborhood selection. We conclude our work by demonstrating our models on simulated networks and by learning a breast cancer microRNA expression network form count-valued next generation sequencing data.

## 2 Poisson Graphical Models & Truncation

Poisson graphical models were introduced by [17] for the pairwise case, where they termed these "Poisson auto-models"; [18, 19] provide a generalization to these models. Let $X = (X_1, X_2, \ldots, X_p)$ be a $p$-dimensional random vector where the domain $\mathcal{X}$ of each $X_s$ is $\{0, 1, 2, \ldots\}$; and let $G = (V, E)$ be an undirected graph over $p$ nodes corresponding to the $p$ variables. The pairwise Poisson graphical model (PGM) distribution over $X$ is then defined as

$$P(X) = \exp\left\{ \sum_{s \in V} (\theta_s X_s - \log(X_s!)) + \sum_{(s,t) \in E} \theta_{st} X_s X_t - A(\theta) \right\}. \qquad (1)$$

It can be seen that the node-conditional distributions for the above distribution are given by $P(X_s | X_{V \setminus s}) = \exp\{\eta_s X_s - \log(X_s!) - \exp(\eta_s)\}$, which is a univariate Poisson distribution with parameter $\lambda = \exp(\eta_s) = \exp(\theta_s + \sum_{t \in \mathcal{N}(s)} \theta_{st} X_t)$, and where $\mathcal{N}(s)$ is the neighborhood of node $s$ according to graph $G$.

As we have noted, there is a major drawback with this Poisson graphical model distribution. Note that the domain of parameters $\theta$ of the distribution in (1) are specified by the normalizability condition $A(\theta) < +\infty$, where $A(\theta) := \log \sum_{\mathcal{X}^p} \exp\left\{ \sum_{s \in V}(\theta_s X_s - \log(X_s!)) + \sum_{(s,t) \in E} \theta_{st} X_s X_t \right\}$.

**Proposition 1** (See [17]). *Consider the Poisson graphical model distribution in* (1). *Then, for any parameter $\theta$, $A(\theta) < +\infty$ only if the pairwise parameters are non-positive: $\theta_{st} \leq 0$ for $(s,t) \in E$.*

The above proposition asserts that the Poisson graphical model in (1) only allows negative edge-weights, and consequently can only capture *negative conditional relationships* between variables. Thus, even though the Poisson graphical model is a natural extension of the univariate Poisson distribution, it entails a highly restrictive parameter space with severely limited applicability. The objective of this paper, then, is to arrive at a graphical model for count data that would allow relaxing these restrictive assumptions, and model both positively and negatively correlated variables.

### 2.1 Truncation, Winsorization, and the Poisson Distribution

The need for finiteness of $A(\theta)$ imposes a negativity constraint on $\theta$ because of the countably infinite domain of the random variables. A natural approach to address this would then be to *truncate* the domain of the Poisson random variables. In this section, we will investigate the two natural ways in which to do so and discuss their possible graphical model distributions.

#### 2.1.1 A Natural Truncation Approach

Kaiser and Cressie [20] first introduced an approach to truncate the Poisson distribution in the context of graphical models. Suppose $Z'$ is Poisson with parameter $\lambda$. Then, one can define what they termed a *Winsorized* Poisson random variable $Z$ as follows: $Z = \mathbb{I}(Z' < R)Z' + \mathbb{I}(Z' \geq R)R$, where $\mathbb{I}(A)$ is an indicator function, and $R$ is a fixed positive constant denoting the truncation level. The probability mass function of this truncated Poisson variable, $P(Z; \lambda, R)$, can then be written as $\mathbb{I}(Z < R)\left(\frac{\lambda^Z}{Z!} \exp(-\lambda)\right) + \mathbb{I}(Z = R)\left(1 - \sum_{i=0}^{R-1} \frac{\lambda^i}{i!} \exp(-\lambda)\right)$. Now consider the use of this Winsorized Poisson distribution for *node-conditional* distributions, $P(X_s | X_{V \setminus s})$: $\mathbb{I}(X_s < R)\left(\frac{\lambda_s^{X_s}}{X_s!} \exp(-\lambda_s)\right) + \mathbb{I}(X_s = R)\left(1 - \sum_{k=0}^{R-1} \frac{\lambda_s^k}{k!} \exp(-\lambda_s)\right)$, where $\lambda_s = \exp(\eta_s) = \exp\left(\theta_s + \sum_{t \in \mathcal{N}(s)} \theta_{st} X_t\right)$. By the Taylor series expansion of the exponential function, this distribution can be expressed in a form *reminiscent* of the exponential family,

$$P(X_s | X_{V \setminus s}) = \exp\left\{ \eta_s X_s - \log(X_s!) + \mathbb{I}(X_s = R)\Psi(\eta_s) - \exp(\eta_s) \right\}, \qquad (2)$$

where $\Psi(\eta_s)$ is defined as $\log\left\{ \frac{R!}{\exp(R\eta_s)} \sum_{k=R}^{\infty} \frac{\exp(k\eta_s)}{k!} \right\}$.

We now have the machinery to describe the development in [20] of a Winsorized Poisson graphical model. Specifically, Kaiser and Cressie [20] assert in a Proposition of their paper that there is a valid *joint distribution* consistent with these Winsorized Poisson node-conditional distributions above. However, in the following theorem, we prove that such a joint distribution *can never exist*.

**Theorem 1.** *Suppose* $X = (X_1, \ldots, X_p)$ *is a p-dimensional random vector with domain* $\{0, 1, ..., R\}^p$ *where* $R > 3$. *Then there is* **no** *joint distribution over* $X$ *such that the corresponding node-conditional distributions* $P(X_s | X_{V \setminus s})$, *of a node conditioned on the rest of the nodes, have the form specified as* $P(X_s | X_{V \setminus s}) \propto \exp\left\{E(X_{V \setminus s})X_s - \log(X_s!) + \mathbb{I}(X_s = R)\Psi\big(E(X_{V \setminus s})\big)\right\}$, *where* $E(X_{V \setminus s})$, *the canonical exponential family parameter, can be an arbitrary function.*

Theorem 1 thus shows that we cannot just substitute the Winsorized Poisson distribution in the construction of [17, 18, 19] to obtain a *Winsorized* variant of Poisson graphical models.

### 2.1.2 A New Approach to Truncation

It is instructive to study the probability mass function of the univariate Winsorized Poisson distribution in (2). The "remnant" probability mass of the Poisson distribution for the cases where $X > R$, was all moved to $X = R$. In the process, it is no longer an exponential family, a property that is crucial for compatibility with the construction in [17, 18, 19]. Could we then derive a truncated Poisson distribution that still belongs to the exponential family? It can be seen that the following distribution over a truncated Poisson variable $Z \in \mathcal{X} = \{0, 1, \ldots, R\}$ fits the bill perfectly: $P(Z) = \frac{\exp\{\theta Z - \log(Z!)\}}{\sum_{k \in \mathcal{X}} \exp\{\theta k - \log(k!)\}}$. The random variable $Z$ here is another natural truncated Poisson variant, where the "remnant" probability mass for the cases where $X > R$ was distributed to all the remaining events $X \leq R$. It can be seen that this distribution also belongs to the exponential family. A natural strategy would then be to use this distribution as the node-conditional distributions in the construction of [17, 18]:

$$P(X_s | X_{V \setminus s}) = \frac{\exp\left\{\left(\theta_s + \sum_{t \in \mathcal{N}(s)} \theta_{st} X_t\right) X_s - \log(X_s!)\right\}}{\sum_{k \in \mathcal{X}} \exp\left\{\left(\theta_s + \sum_{t \in \mathcal{N}(s)} \theta_{st} X_t\right) k - \log(k!)\right\}} . \qquad (3)$$

**Theorem 2.** *Suppose* $X = (X_1, X_2, \ldots, X_p)$ *be a p-dimensional random vector, where each variable* $X_s$ *for* $s \in V$ *takes values in the truncated positive integer set,* $\{0, 1, ..., R\}$, *where* $R$ *is a fixed positive constant. Suppose its node-conditional distributions are specified as in* (3), *where the node-neighborhoods are as specified by a graph* $G$. *Then, there exists a unique joint distribution that is consistent with these node-conditional distributions, and moreover this distribution belongs to the graphical model represented by* $G$, *with the form:* $P(X) := \exp\left\{\sum_{s \in V}(\theta_s X_s - \log(X_s!)) + \sum_{(s,t) \in E} \theta_{st} X_s X_t - A(\theta)\right\}$, *where* $A(\theta)$ *is the normalization constant.*

We call this distribution the Truncated Poisson graphical model (TPGM) distribution. Note that it is distinct from the original Poisson distribution (1); in particular its normalization constant involves a summation over finitely many terms. Thus, no restrictions are imposed on the parameters for the normalizability of the distribution. Unlike the original Poisson graphical model, the TPGM can model both positive and negative dependencies among its variables.

There are, however, some drawbacks to this graphical model distribution. First, the domain of the variables is bounded a priori by the distribution specification, so that it is not broadly applicable to arbitrary, and possibly infinite, count-valued data. Second, problems arise when the random variables take on large count values close to $R$. In particular by examining (3), one can see that when $X_t$ is large, the mass over $X_s$ values get pushed towards $R$; thus, this truncated version is not always close to that of the original Poisson density. Therefore, as the truncation value $R$ increases, the possible values that the parameters $\theta$ can take become increasingly negative or close to zero to prevent all random variables from always taking large count values at the same time. This can be seen as if we take $R \to \infty$, we arrive at the original PGM and negativity constraints. In summary, the TPGM approach offers some trade-offs between the value of $R$, it more closely follows the Poisson density when $R$ is large, and the types of dependencies permitted.

## 3 A New Class of Poisson Variants and Their Graphical Model Distributions

As discussed in the previous section, taking a Poisson random variable and truncating it may be a natural approach but does not lead to a valid multivariate graphical model extension, or does so with some caveats. Accordingly in this section, we investigate the possibility of modifying the Poisson distribution more fundamentally, by modifying its sufficient statistic and base measure.

Let us first briefly review the derivation of a Poisson graphical model as the graphical model extension of a univariate exponential family distribution from [17, 18, 19]. Consider a general univariate exponential family distribution, for a random variable $Z$: $P(Z) = \exp(\theta B(Z) - C(Z) - D(\theta))$, where $B(Z)$ is the exponential family sufficient statistic, $\theta \in \mathbb{R}$ is the parameter, $C(Z)$ is the base measure, and $D(\theta)$ is the log-partition function. Suppose the node-conditional distributions are all specified by the above exponential family,

$$P(X_s|X_{V \setminus s}) = \exp\{E(X_{V \setminus s})\, B(X_s) + C(X_s) - \bar{D}(X_{V \setminus s})\}, \tag{4}$$

where the canonical parameter of exponential family is some function $E(\cdot)$ on the rest of the variables $X_{V \setminus s}$ (and hence so is the log-normalization constant $\bar{D}(\cdot)$). Further, suppose the corresponding joint distribution factors according to the graph $G$, with the factors over cliques of size at most $k$. Then, Proposition 2 in [18], shows that there exists a unique joint distribution corresponding to the node-conditional distributions in (4). With clique factors of size $k$ at most two, this joint distribution takes the following form: $P(X) = \exp\big\{ \sum_{s \in V} \theta_s B(X_s) + \sum_{(s,t) \in E} \theta_{st}\, B(X_s)B(X_t) - \sum_{s \in V} C(X_s) - A(\theta) \big\}$. Note that although the log partition function $A(\theta)$ is usually computationally intractable, the log-partition function $\bar{D}(\cdot)$ of its node-conditional distribution (4) is still tractable, which allows consistent graph structure recovery [18]. Also note that the original Poisson graphical model (1) discussed in Section 2 can be derived from this construction with sufficient statistics $B(X) = X$, and base measure $C(X) = \log(X!)$.

### 3.1 A Quadratic Poisson Graphical Model

As noted in Proposition 1, the normalizability of this Poisson graphical model distribution, however, requires that the pairwise parameters be negative. A closer look at the proof of Proposition 1 shows that a key driver of the result is that the base measure terms $\sum_{s \in V} C(X_s) = \sum_{s \in V} \log(X_s!)$ scale more slowly than the quadratic pairwise terms $X_s X_t$. Accordingly, we consider the following general distribution over count-valued variables:

$$P(Z) = \exp(\theta Z - C(Z) - D(\theta)), \tag{5}$$

which has the same sufficient statistics as the Poisson, but a more general base measure $C(Z)$, for some function $C(\cdot)$. The following theorem shows that for normalizability of the resulting graphical model distribution with possibly positive edge-parameters, the base measure cannot be sub-quadratic:

**Theorem 3.** *Suppose $X = (X_1, \ldots, X_p)$ is a count-valued random vector, with joint distribution given by the graphical model extension of the univariate distribution in* (5) *which follows the construction of [17, 18, 19]). Then, if the distribution is normalizable so that $A(\theta) < \infty$ for $\theta \nleq 0$, it necessarily holds that $C(Z) = \Omega(Z^2)$.*

The previous theorem thus suggests using the "Gaussian-esque" quadratic base measure $C(Z) = Z^2$, so that we would obtain the following distribution over count-valued vectors, $P(X) = \exp\big\{ \sum_{s \in V} \theta_s X_s + \sum_{(s,t) \in E} \theta_{st} X_s X_t - c \sum_{s \in V} X_s^2 - A(\theta) \big\}$. for some fixed positive constant $c > 0$. We consider the following generalization of the above distribution:

$$P(X) = \exp\left\{ \sum_{s \in V} \theta_s X_s + \sum_{(s,t) \in E} \theta_{st}\, X_s X_t + \sum_{s \in V} \theta_{ss} X_s^2 - A(\theta) \right\}. \tag{6}$$

We call this distribution the Quadratic Poisson Graphical Model (QPGM). The following proposition shows that the QPGM is normalizable while permitting both positive and negative edge-parameters.

**Proposition 2.** *Consider the distribution in* (6). *Suppose we collate the quadratic term parameters into a $p \times p$ matrix $\Theta$. Then the distribution is normalizable provided the following condition holds: There exists a positive constant $c_\theta$, such that for all $X \in \mathbf{W}^p$, $X^T \Theta X \leq -c_\theta \|X\|_2^2$.*

The condition in the proposition would be satisfied provided that the pairwise parameters are pointwise negative: $\Theta < 0$, similar to the original Poisson graphical model. Alternatively, it is also sufficient for the pairwise parameter matrix to be negative-definite: $\Theta \prec 0$, which does allow for positive and negative dependencies, as in the Gaussian distribution.

A possible drawback with this distribution is that due to the quadratic base measure, the QPGM has a Gaussian-esque thin tail. Even though the domains of Gaussian and QPGM are distinct,

their densities have similar behaviors and shapes as long as $\theta_s + \sum_{t\in N(s)} \theta_{st} X_t \geq 0$. Indeed, the Gaussian log-partition function serves as a variational upper bound for the QPGM. Specifically, under the restriction that $\theta_{ss} < 0$, we arrive at the following upper bound:

$$D(\theta; X_{V\setminus s}) = \log \sum_{X_s \in \mathbf{W}} \exp\left\{\eta_s X_s + \theta_{ss} X_s^2\right\} \leq \log \int_{X_s \in \mathbf{R}} \exp\left\{\eta_s X_s + \theta_{ss} X_s^2\right\} dX_s$$

$$= D_{\text{Gauss}}(\theta; X_{\setminus s}) = 1/2 \log 2\pi - 1/2 \log(-2\theta_{ss}) - \frac{1}{4\theta_{ss}}\left(\theta_s + \sum_{t\in N(s)} \theta_{st} X_t\right)^2,$$

by relating to the log-partition function of a node-conditional Gaussian distribution. Thus, node-wise regressions according to the QPGM via the above variational upper bound on the partition function would behave similarly to that of a Gaussian graphical model.

### 3.2 A Sub-Linear Poisson Graphical Model

From the previous section, we have learned that so long as we have linear sufficient statistics, $B(X) = X$, we must have a base measure that scales at least quadratically, $C(Z) = \Omega(Z^2)$, for a Poisson-based graphical model (i) to permit both positive and negative conditional dependencies and (ii) to ensure normalizability. Such a quadratic base measure however results in a Gaussian-esque thin tail, while we would like to specify a distribution with possibly heavier tails than those of QPGM. It thus follows that we would need to control the linear Poisson sufficient statistics $B(X) = X$ itself. Accordingly, we consider the following univariate distribution over count-valued variables:

$$P(Z) = \exp(\theta B(Z; R_0, R) - \log Z! - D(\theta, R_0, R)), \tag{7}$$

which has the same base measure $C(Z) = \log Z!$ as the Poisson, but with the following sub-linear sufficient statistics:

$$B(x; R_0, R) = \begin{cases} x & \text{if } x \leq R_0 \\ -\frac{1}{2(R-R_0)}\, x^2 + \frac{R}{R-R_0}\, x - \frac{R_0^2}{2(R-R_0)} & \text{if } R_0 < x \leq R \\ \frac{R+R_0}{2} & \text{if } x \geq R \end{cases}$$

We depict this sublinear statistic in Figure 3 in the appendix; Up to $R_0$, $B(x)$ increases linearly, however, after $R_0$ its slope decreases linearly and becomes zero at $R$.

The following theorem shows the normalizability of the SPGM:

**Theorem 4.** *Suppose $X = (X_1, \ldots, X_p)$ is a count-valued random vector, with joint distribution given by the graphical model extension of the univariate distribution in* (7) *(following the construction [17, 18, 19]):*

$$P(X) = \exp\left\{ \sum_{s\in V} \theta_s B(X_s; R_0, R) + \sum_{(s,t)\in E} \theta_{st}\, B(X_s; R_0, R) B(X_t; R_0, R) - \sum_{s\in V} \log(X_s!) - A(\theta, R_0, R) \right\}.$$

*This distribution is normalizable, so that $A(\theta) < \infty$ for all pairwise parameters $\theta_{st} \in \mathbb{R}; (s,t) \in E$.*

On comparing with the QPGM, the SPGM has two distinct advantages: (1) it has a heavier tails with milder base measures as seen in its motivation, and (2) allows a broader set of feasible pairwise parameters (actually for all real values) as shown in Theorem 4.

The log-partition function $D(\theta, R_0, R)$ of node-conditional SPGM involves the summation over infinite terms, and hence usually does not have a closed-form. The log-partition function of traditional univariate Poisson distribution, however, can serve as a variational upper bound:

**Proposition 3.** *Consider the node-wise conditional distributions in* (7)*. If $\theta \geq 0$, we obtain the following upper bound:*

$$D(\theta, R_0, R) \leq D_{Pois}(\theta) = \exp(\theta).$$

## 4   Numerical Experiments

While the focus of this paper is model specification, we can learn our models from iid samples of count-valued multivariate vectors using neighborhood selection approaches as suggested in [1, 5,

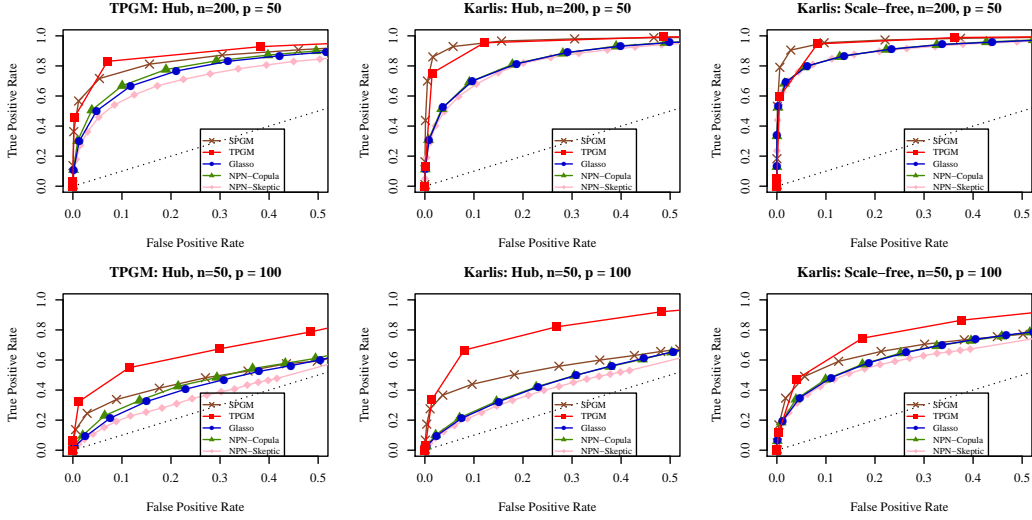

Figure 1: ROC curves for recovering the true network structure of count-data generated by the TPGM distribution or by [15] (sums of independent Poissons method) for both standard and high-dimensional regimes. Our TPGM and SPGM $M$-estimators are compared to the graphical lasso [4], the non-paranormal copula-based method [7] and the non-paranormal SKEPTIC estimator [10].

6, 18]. Specifically, we maximize the $\ell_1$ penalized node-conditional likelihoods for our TPGM, QPGM and SPGM models using proximal gradient ascent. Also, as our models are constructed in the framework of [18, 19], we expect extensions of their sparsistency analysis to confirm that the network structure of our model can indeed be learned from iid data; due to space limitations, this is left for future work.

**Simulation Studies.** We evaluate the comparative performance of our TPGM and SPGM methods for recovering the true network from multivariate count data. Data of dimension $n = 200$ samples and $p = 50$ variables or the high-dimensional regime of $n = 50$ samples and $p = 100$ variables is generated via the TPGM distribution using Gibbs sampling or via the sums of independent Poissons method of [15]. For the former, edges were generated with both positive and negative weights, while for the latter, only edges with positive weights can be generated. As we expect the SPGM to be sparsistent for data generated from the SGPM distribution following the work of [18, 19], we have chosen to present results for data generated from other models. Two network structures are considered that are commonly used throughout genomics: the hub and scale-free graph structures. We compare the performance of our TPGM and SPGM methods with $R$ set to the maximum count value to Gaussian graphical models [4], the non-paranormal [7], and the non-paranormal SKEPTIC [10].

In Figure 1, ROC curves computed by varying the regularization parameter, and averaged over 50 replicates are presented for each scenario. Both TPGM and SPGM have superior performance for count-valued data than Gaussian based methods. As expected, the TPGM method has the best results when data is generated according to its distribution. Additionally, TPGM shows some advantages in high-dimensional settings. This likely results from a facet of its node-conditional distribution which places larger mass on strongly dependent count values that are close to $R$. Thus, the TPGM method may be better able to infer edges from highly connected networks, such as those considered. Additionally, all methods compared outperform the original Poisson graphical model estimator, given in [18] (results not shown), as this method can only recover edges with negative weights.

**Case Study: Breast Cancer microRNA Networks.** We demonstrate the advantages of our graphical models for count-valued data by learning a microRNA (miRNA) expression network from next generation sequencing data. This data consists of counts of sequencing reads mapped back to a reference genome and are replacing microarrays, for which GGMs are a popular tool, as the preferred measures of gene expression [22]. Level III data was obtained from the Cancer Genome Atlas (TCGA) [23] and processed according to techniques described in [24]; this data consists of $n = 544$ subjects and $p = 262$ miRNAs. Note that [18, 24] used this same data set to demonstrate

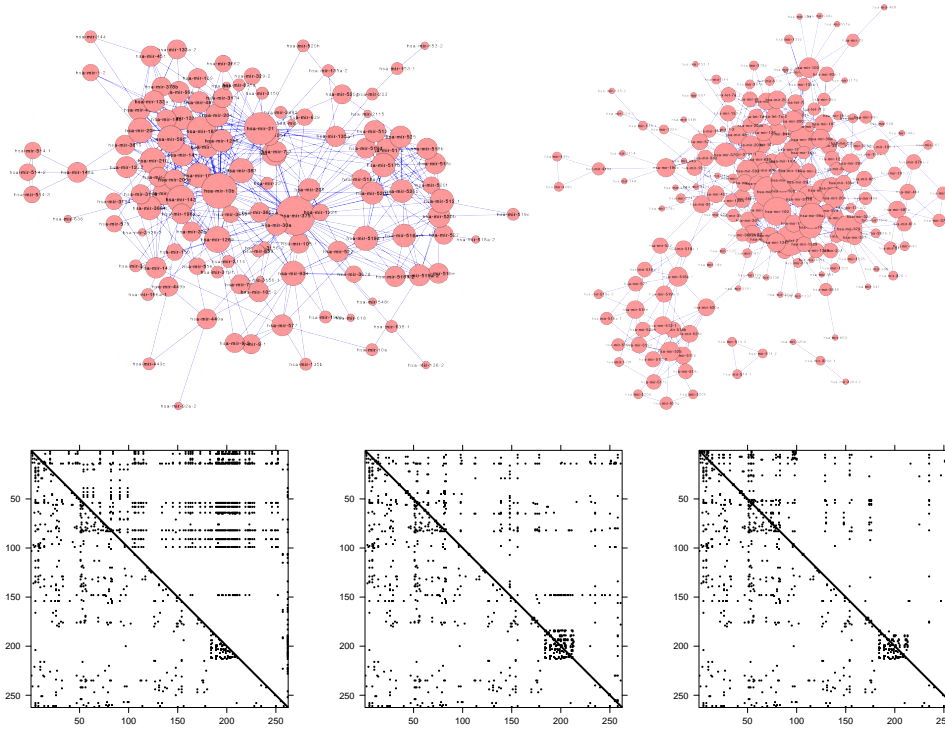

Figure 2: Breast cancer miRNA networks. Network inferred by (top left) TPGM with $R = 11$ and by (top right) SPGM with $R = 11$ and $R_0 = 5$. The bottom row presents adjacency matrices of inferred networks with that of SPGM occupying the lower triangular portion and that of (left) PGM, (middle) TPGM with $R = 11$, and graphical lasso (right) occupying the upper triangular portion.

network approaches for count-data, and thus, we use the same data set so that the results of our novel methods may be compared to those of existing approaches.

Networks were learned from this data using the original Poisson graphical model, Gaussian graphical models, our novel TPGM approach with $R = 11$, the maximum count, and our novel SPGM approach with $R = 11$ and $R_0 = 5$. Stability selection [25] was used to estimate the sparsity of the networks in a data-driven manner. Figure 2 depicts the inferred networks for our TPGM and SPGM methods as well as comparative adjacency matrices to illustrate the differences between our SPGM method and other approaches. Notice that SPGM and TPGM find similar network structures, but TPGM seems to find more hub miRNAs. This is consistent with the behavior of the TPGM distribution when strongly correlated counts have values close to $R$. The original Poisson graphical model, on the other hand, misses much of the structure learned by the other methods and instead only finds 14 miRNAs that have major conditionally negative relationships. As most miRNAs work in groups to regulate gene expression, this result is expected and illustrates a fundamental flaw of the PGM approach. Compared with Gaussian graphical models, our novel methods for count-valued data find many more edges and biologically important hub miRNAs. Two of these, mir-375 and mir-10b, found by both TPGM and SPGM but not by GGM, are known to be key players in breast cancer [26, 27]. Additionally, our TPGM and SPGM methods find a major clique which consists of miRNAs on chromosome 19, indicating that this miRNA cluster may by functionally associated with breast cancer.

## Acknowledgments

The authors acknowledge support from the following sources: ARO via W911NF-12-1-0390 and NSF via IIS-1149803 and DMS-1264033 to E.Y. and P.R; Ken Kennedy Institute for Information Technology at Rice to G.A. and Z.L.; NSF DMS-1264058 and DMS-1209017 to G.A.; and NSF DMS-1263932 to Z.L..

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
