[Supplementary Material]

# Appendix

## A  Discussion

Our work for the first time has provided a graphical model distribution for high-dimensional count-valued data that permits general dependencies between variables. We have shown that the PGM of [17] can only capture negative conditional dependencies and the Winsorization of [20] can never lead to a proper joint graphical model distribution. Our novel TPGM uses an alternative approach to truncation permits a proper joint density, but with several drawbacks. To address these, we have investigated alterations to the base measure and sufficient statistics of the univariate Poisson distribution, leading to our novel QPGM and SPGM approaches. The latter uses sub-linear sufficient statistics to mitigate the effect of large counts, thus permitting both positive and negative conditional dependencies. This paper presents a thorough investigation of Poisson Graphical Model specification, from which we can conclude that it is indeed possible to specify a widely applicable graphical model for high-dimensional count data.

There are many future items for further research related to our work. We have briefly described a possible approach to sparse graph estimation according to our model by penalized neighborhood selection. Specific algorithms for fitting our models, including possible variational approaches will be investigated in future work. In particular, it may be of interest to consider the Local PGM proposed in [24] as a special case of the TPGM, QPGM, and SPGM, specifically providing an upper bound for the log-partition function of the latter. Such approaches and the statistical recovery properties of these methods, including consistent graph recovery, are avenues for future research.

## B  Proofs

### B.1  Proof of Proposition 1

Suppose $\theta_{st} > 0$ for any $(s,t) \in E$. Then, recalling Stirling's formula: $\ln(n!) = n\ln(n) - n + O(\ln n)$, it can be seen that

$$\theta_{st}x_sx_t + \theta_sx_s + \theta_tx_t - \ln x_s! - \ln x_t!$$
$$= \theta_{st}x_sx_t + \theta_sx_s + \theta_tx_t - x_s\ln x_s - x_t\ln x_t + x_s + y_s + O(\ln x_s + \ln x_t)$$
$$\to \infty, \quad \text{as } x_s, x_t \to \infty,$$

which would result in the distribution in (1) not being normalizable; so that it follows that $\theta_{st} \le 0$. The statement of the proposition follows.

### B.2  Proof of Theorem 1

We will prove by contradiction.

Following the notations in [17, 18, 19], we denote $Q(X)$ as

$$Q(X) = \log(P(X)/P(\mathbf{0})),$$

Figure 3: Sublinear sufficient statistics, $B(X; R_0, R)$

for any $X = (X_1, ..., X_p) \in \{0, 1, ..., R\}^p$. In this proof, we are going to focus only on the pairwise MRF, however note that even with the higher order dependencies, the statement holds since the pairwise terms satisfying the condition of the theorem do not exist:

$$Q(X) = \sum_{s \in V} X_s G_s(X_s) + \sum_{(s,t) \in E} X_s X_t G_{st}(X_s, X_t). \tag{8}$$

In order to specify the joint distribution, we need to compute the function $G_s$ and $G_{st}$ in (??).

It is useful to consider the relationship between the function $Q(X)$, and the conditional distribution $P(X_s | X_{N(s)})$:

$$\exp(Q(X) - Q(\bar{X}_s)) = P(X)/P(\bar{X}_s) \tag{9}$$
$$= P(X_s | X_{N(s)})/P(0 | X_{N(s)}),$$

where $\bar{X}_s := (X_1, \ldots, X_{s-1}, 0, X_{s+1}, \ldots, X_p)$. We then obtain

$$X_s G_s(X_s) + X_s \sum_{t \in N(s)} X_t G_{st}(X_s, X_t) = -\log(X_s!) +$$
$$E(X_{V \setminus s})X_s + \mathbb{I}(X_s = R)\Psi\big(E(X_{V \setminus s})\big) \tag{10}$$

We can obtain the first order function $X_s G_s(X_s)$ by setting $X_t = 0$ for all $t \neq s$ in (??):

$$X_s G_s(X_s) = E(\mathbf{0})X_s + \mathbb{I}(X_s = R)\Psi\big(E(\mathbf{0})\big) - \log(X_s!). \tag{11}$$

Suppose nodes $s$ and $t$ are neighbors, i.e. $\theta_{st} \neq 0$. Setting $X_r = 0$ for all $r \notin \{s, t\}$, we obtain

$$X_s G_s(X_s) + X_s X_t G_{st}(X_s, X_t) \tag{12}$$
$$= E(0, \ldots, X_t, \ldots, 0)X_s + \mathbb{I}(X_s = R)\Psi\big(E(0, \ldots, X_t, \ldots, 0)\big) - \log(X_s!).$$

Combining (??) and (??) yields

$$X_s X_t G_{st}(X_s, X_t) =$$
$$\Big\{E(0, \ldots, X_t, \ldots, 0) - E(\mathbf{0})\Big\}X_s + \mathbb{I}(X_s = R)\Big\{\Psi\big(E(0, \ldots, X_t, \ldots, 0)\big) - \Psi\big(E(\mathbf{0})\big)\Big\}. \tag{13}$$

Similarly, considering the difference of $Q$ values of $X$ and $\bar{X}_t$ in (??), we obtain

$$X_s X_t G_{st}(X_s, X_t) =$$
$$\Big\{E(0, \ldots, X_s, \ldots, 0) - E(\mathbf{0})\Big\}X_t + \mathbb{I}(X_t = R)\Big\{\Psi\big(E(0, \ldots, X_s, \ldots, 0)\big) - \Psi\big(E(\mathbf{0})\big)\Big\}. \tag{14}$$

Note that (??) and (??) should be the same for all possible pairs of $X_s$ and $X_t$. We first consider the case where $X_s, X_t \in \{1, \ldots, R-1\}$. Then, the indicator functions will fail to satisfy the arguments and disappear. In this case, as shown in [18, 19], we can simply deduce

$$E(0, \ldots, X_s, \ldots, 0) - E(\mathbf{0}) = \beta_{st} X_s \quad \text{if } X_s \in \{1, \ldots, R-1\} \tag{15}$$

where $\beta_{st}$ is some constant.

Now, we fix $X_t = R$ and again $X_s \in \{1, R-1\}$. Then, we can combine (??) and (??) with one indicator function out of two:

$$\Big\{E(0, \ldots, X_t = R, \ldots, 0) - E(\mathbf{0})\Big\}X_s =$$
$$\Big\{E(0, \ldots, X_s, \ldots, 0) - E(\mathbf{0})\Big\}R + \Psi\big(E(0, \ldots, X_s, \ldots, 0)\big) - \Psi\big(E(\mathbf{0})\big).$$

Since we are assuming $X_s \in \{1, \ldots, R-1\}$, by (??), some elementary algebra yields that

$$c_1 X_s = \Psi(c_2 X_s) + c_3.$$

where $c_1, c_2$ and $c_3$ are all fixed constants with respect to $X_s$.

Since the slope of function $\Psi(c_2 X_s)$ is strictly increasing (i.e. $\Psi''(c_2 X_s) > 0$) by definition, $\Psi(c_2 X_s)$ cannot be same as a constant factor of $X_s$ at more than 2 values in $\{1, \ldots, R-1\}$.

### B.3 Proof of Theorem 4

If $\theta_{xy} \leq 0$, it can be trivially shown that the probability mass function is normalizable by similar reasoning as in Proposition 1. Given parameters $\theta_{xy} > 0$, $\theta_x$ and $\theta_y$, consider some positive integer $a$ that is large enough to satisfy $\theta_{xy}R^2 + |\theta_x|R + |\theta_y|R \leq |\theta_x|a$. Similarly, also consider some positive integer $b$ s.t. $\theta_{xy}R^2 + |\theta_x|R + |\theta_y|R \leq |\theta_y|b$. Then, for all $(x, y)$ s.t. $x \geq a$ or $y \geq b$, we have $\theta_{xy}R^2 + |\theta_x|R + |\theta_y|R \leq |\theta_x|x + |\theta_y|y$. If each entry of one sequence is smaller than that of another sequence, its summation is also smaller. Therefore, we have

$$\sum_{x \geq a \text{ or } y \geq b} \exp\left(\theta_{xy}B(x)B(y) + \theta_x B(x) + \theta_y B(y) - \log x! - \log y!\right)$$

$$\overset{(i)}{\leq} \sum_{x \geq a \text{ or } y \geq b} \exp\left(\theta_{xy}R^2 + |\theta_x|R + |\theta_y|R - \log x! - \log y!\right)$$

$$\leq \sum_{x \geq a \text{ or } y \geq b} \exp\left(|\theta_x|x + |\theta_y|y - \log x! - \log y!\right),$$

where in inequality $(i)$ we use the fact that $\theta_{xy} > 0$. Since $\sum_{x \geq a \text{ or } y \geq b} \exp\left(|\theta_x|x + |\theta_y|y - \log x! - \log y!\right)$ is normalizable, so is $\sum_{x \geq a \text{ or } y \geq b} \exp\left(\theta_{xy}B(x)B(y) + \theta_x B(x) + \theta_y B(y) - \log x! - \log y!\right)$, which completes the proof.

### B.4 Proof of Proposition 3

$$D(\theta, R_0, R) = \log \sum_{Z \in \mathbf{W}} \exp\left\{\theta B(Z) - \log(Z!)\right\}$$

$$\leq \log \sum_{Z \in \mathbf{W}} \exp\left\{\theta Z - \log(Z!)\right\} = D_{\text{Pois}}(\theta) = \exp(\theta)$$

where the inequality holds if $\theta \geq 0$.