[Reviews · NeurIPS 2013]

Submitted by Assigned_Reviewer_3

The paper describes a number of new models for representing a joint distribution over integer-count variables. The authors argue that the "default" model that arises from Yang et al. is not satisfactory because it can only model negative correlations in order for the distribution to be normalized. They then consider a series of fixes for this including a new truncation method, using a quadratic base measure statistic (which they prove is necessary with everything else fixed), and finally a sub-linear sufficient statistic. The authors present experimental results on both synthetic and real data.

This is a well written paper describing some nice solutions for representing count data. I believe this is original, and the results are significant.

I found the "big negatives" of the truncation and quadratic models to be somewhat unconvincing. For the truncation, I could imagine that there would be a lot of interesting domains for which counts typically don't get too high and the model would work well. For the quadratic model, couldn't "Gaussian-esque thin tails" be appropriate sometimes?

I don't know what "sparsistent" means, but I see that this term has been used for a while.

Page 3, above 2.1.1: "graphical mode distributions" => "model"
Page 5, above 3.1: "Note that the log..." Do you mean "Note that although the log"?
Summary: Well written quality paper on modeling a joint distribution over count variables.

Submitted by Assigned_Reviewer_6

The goal of constructing multivariate distributions that are
appropriate for count data is an important one, and the authors
correctly identify limitations of current approaches (a reservation on
this is noted below).

On the theoretical and conceptual front, the paper has several
merits. Starting with a proof that previously suggested truncation
does not lead to a valid model, the authors suggest several alternatives.
The TPGM truncation, though obvious, is warranted due to its natural
interpretation. The QPGM and SPGM, while technically simple, are
novel in that the base measure and sufficient statistics of the
exponential representation are put pm the table as the means to
modify the Poisson distribution. Further, the changes made are
well motivated and appealing. That said, the actual theory involved
follows almost directly from the work of Yang et al so that we are
left with several possible model suggestions whose merit need to be
evaluated in practice.

In light of the above, the experimental evaluation is disappointing,
particularly since the declared goal of the paper is to fix the practical
limitations of Poisson graphical models. While the synthetic experiment
shows some potential, the data is geared toward the models suggested.
In the real experiments, the comparison to copula-based alternatives is
glaringly missing. Further, the reported results are only qualitatively anecdotal.
Since all method involve the construction of joint distributions, a more
objective log-probability of test data evaluation is needed.

More important, and indeed this touches on the core issue of whether
there is a need for the new model, is the fact that the copula competitors
are used in an overly black-box manner. In particular, even when using
only a Gaussian copula, it makes more sense to use some sensible marginal
model rather than the one used in the non-paranormal (and that has no density!)
which was mainly chosen due to its asymptotic properties rather than
practical merits. The essentially identical performance to Glasso is suspicious
and I strongly suspect that even simple Gaussian kernel density
estimates would do much better. Similarly, if applied to the real-data,
the (sensible) choice of using R=11 can also be translated to a
choice for the marginal of the copula. I do not expect in-depth exploration
here but some reasonable baseline is warranted.

Finally, the paper is generally well written. Though I believe all results
are true, particularly since the authors start with an error in another
work, I suggest including all proofs in the supplementary material.
Also, I felt that the end of section 3.1 was overly detailed and that
the bound did not contribute to the Gaussian-esque argument.
On the other hand, I would take 3.2 more slowly as it is the heart
of the suggested method and in particular not defer the figure to the
supplementary material but rather present it and better explain its intuition.
Summary: Based on a conditional exponential construction,
the authors present alternatives to the
Poisson graphical model with the goal of allowing for
flexible joint modeling with a mix of positive and
negative dependencies.

On the good side, the approach suggested is appealing
and has some theoretical novelty. On the bad side, the
experimental evaluation is limited and somewhat biased
so that the bottom line is yet another multivariate Poisson-like
model whose merit is unclear.

Submitted by Assigned_Reviewer_8

The paper provides a construction for multivariate distributions over
unbounded counts that obey the Markov structure of an undirected
network.

Building a multivariate distribution over unbounded counts is in
general a hard problem, as studied at length by Besag and others for
the past 30 years (including textbooks such as Arnold et al.'s
"Conditional Specification of Statistical Models", 1999, which extends
some of the observations of the authors to other distributions such as
conditionally-specified exponential distributions).

In the end, the proposal given by the authors succeeds in some
relevant ways. The upsides are constructions that do allow for
marginal distributions over counts and which lead to relatively simple
estimation algorithms. The downsides are, QPGM has marginally thinner
tails and SPGM does not have closed-form conditional distributions
(which somehow defeats the point of building a conditionally specified
model). As a matter of fact, I don't even know how SPGM can be called
a Poisson distribution (for QPGM at least one can claim that ``only''
the base measure is being changed). That's OK, but it made me wonder
what the main motivation for modeling counts is, since the Poisson
itself is not a good distribution to fit empirical data anyway (don't
get me wrong, the Poisson is a very useful as a building block to many
models - components of stochastic processes and within latent variable
models etc. - but could you plot your data for breast cancer and tell
me whether it looks anything close to a Poisson?). It would be very
useful to have a plot of the probability mass function of the SPGM
too, which feels somewhat convoluted at first sight. I suppose you are
considering R and R0 as constants (or otherwise these wouldn't be
exponential families). How are they chosen? Which advice do you give
to the practitioner?

That being said: to construct a multivariate distribution over counts
obeying the independence model of MRF is hard. I honestly appreciate
the effort put in this paper and I think the results are of
theoretical interest to NIPS. The only thing that rubs me in the wrong
way is the somewhat overly light appreciation of the literature. For
instance, it almost feels like the authors don't really know what a
copula is. The authors seem not to understand [8] (or at least
definitely presented it in the wrong way), for instance, where the
whole point is to build multivariate distributions for arbitrary
discrete data (count data, inclusive), and for which a battery of MCMC
and approximate inference methods exist. It made me wonder whether the
discussion of [11] truly makes any sense, since the whole point of
that pioneering book is to show how to build discrete models with
log-linear parameters in a way it doesn't grow exponentially with the
number of variables (although fair enough I don't have a copy of it
with me right now and I don't remember anymore what it says about
Poisson distributions). But perhaps the worst omission is a complete
neglect of the vast spatial statistics literature, where
high-dimensional count data analysis has been done for a long time.

It needs to be said that several of these approaches (including the
sparse precision Gaussian copula model of [8]) don't really model
MRF-style independence constraints in the observable space. So as I
said the theoretical contribution of this paper is a valid one. But as
a practitioner I'm not yet convinced why I should pay the price of
sticking to this model space instead of just using the simpler
structured Gaussian random field + Poisson measurement model, which
has been the standard for a long time.

Final comment: I'm not an expert in gene expression analysis at all,
but I would be grateful to have a reference newer than [20] claiming
that ``counts of sequencing reads ... are replacing microarrays''.
Summary: A method for constructing multivariate distributions for counts that is Markov with respect to undirected graphs. Like any nontrivial multivariate construction, it has its advantages and shortcomings. Literature review feels incomplete.
Author Feedback

Author rebuttal: We thank the reviewers for their careful comments and feedback.

Reviewer 3:

> As noted by reviewer, each of our proposed models has its own advantages and shortcomings. Depending on the application at hand, we also believe that TPGM or QPGM might be useful; and indeed we view these as important contributions of the paper as well. As suggested, we will temper some of the negative tone towards TPGM and QPGM in the introduction.

> 'sparsistent' refers to being consistent in recovering the sparsity pattern; i.e. it recovers the sparsity pattern exactly with probability converging to one. We will add a clarification to the text.

> We will fix the typos.

Reviewer 6:

> On synthetic experiments being geared toward our models: we note that two of three columns in Fig. 1 use data generated using the sums of independent Poissons method of Karlis [ref 15]; which does not follow the graphical model machinery in our paper. Note that the plots display six models: three of our proposed models against three other baseline methods (the graphical lasso, the non-paranormal copula-based method, and the non-paranormal SKEPTIC estimator.)
> Comparing against other methods in the real data: we did not show the mRNA network generated using the other copula based methods primarily due to lack of space, but we note moreover that the evaluation in the real data is necessarily qualitative; it is in Fig. 1 that we could perform quantitative comparisons, and where we compared against three other baseline methods.

> Using copulas in a "black box manner": we note that we used the well-cited non-paranormal, which estimates a Gaussian copula using a Winsorized empirical CDF for the marginal distributions, for which they show strong convergence guarantees. Varying the non-paranormal by plugging in other non-parametric estimates for the marginal distributions is certainly an interesting possibility, which we will explore, but we note the nonparametric estimate in the non-paranormal paper had an optimal rate of convergence.

Reviewer 8: The main motivation of SPGM is that we want to keep a mild base measure so that the distribution has heavier tails. We preserve same base measure as in univariate Poisson distribution, and derive the sufficient statistics with which the joint is normalizable. We still call it 'Poisson' in the sense that; for the univariate case, the distribution is well defined even with R=R0=\infty, which is the standard Poisson distribution.

On references: we note that the focus of this work was on graphical model distributions (we note that such graphical model distributions are of great interest in ML due to a long line of recent work on leveraging their algebraic and graphical structure to compute various distribution functionals such as conditional marginals in a computationally efficient manner). Because of this, our presentation of previous work was focused on prior work such as the non-paranormal models which do satisfy Markov independence assumptions. We did not mean to slight other non-graphical-model based work on modeling count data, they were merely outside the scope of this paper. Nonetheless, we will expand upon our literature review in the final version. Our reference to the graphical models book [11] when discussing a long line of work on log-linear models with exponential number of parameters was to point to discussion in [11]; we will clarify this in the final version.